# Non-Robustness of Ang’s Risk Classification in Human Papillomavirus-Related Oropharyngeal Squamous Cell Carcinoma in Japanese Patients

**DOI:** 10.3390/cancers14102442

**Published:** 2022-05-15

**Authors:** Jun Itami, Kenya Kobayashi, Taisuke Mori, Yoshitaka Honma, Yuko Kubo, Naoya Murakami, Go Omura, Kae Okuma, Koji Inaba, Kana Takahashi, Tairo Kashihara, Yuri Shimizu, Ayaka Takahashi, Yuko Nakayama, Fumihiko Matsumoto, Seiichi Yoshimoto, Hiroshi Igaki

**Affiliations:** 1Department of Radiation Oncology, National Cancer Center Hospital, Tokyo 104-0045, Japan; namuraka@ncc.go.jp (N.M.); kokuma@ncc.go.jp (K.O.); koinaba@ncc.go.jp (K.I.); kantakah@ncc.go.jp (K.T.); tkashiha@ncc.go.jp (T.K.); yurshimi@ncc.go.jp (Y.S.); ayakatak@ncc.go.jp (A.T.); yukonak4@ncc.go.jp (Y.N.); hirigaki@ncc.go.jp (H.I.); 2Shin-Matsudo Accuracy Radiation Therapy Center, Shin-Matsudo Central General Hospital, Chiba 270-0034, Japan; 3Department of Head and Neck Surgical Oncology, National Cancer Center Hospital, Tokyo 104-0045, Japan; kenyajp@hotmail.com (K.K.); gomura@ncc.go.jp (G.O.); fmatsu@juntendo.ac.jp (F.M.); seyoshim@ncc.go.jp (S.Y.); 4Department of Pathology, National Cancer Center Hospital, Tokyo 104-0045, Japan; tamori@ncc.go.jp; 5Department of Head and Neck Medical Oncology, National Cancer Center Hospital, Tokyo 104-0045, Japan; yohonma@ncc.go.jp; 6Department of Diagnostic Radiology, National Cancer Center Hospital, Tokyo 104-0045, Japan; yukkubo@ncc.go.jp

**Keywords:** oropharyngeal cancer, squamous cell cancer, human papillomavirus, p16, risk classification, prognosis, Japanese patients

## Abstract

**Simple Summary:**

Human papillomavirus (HPV)-related oropharyngeal squamous cell carcinoma (OPSCC) has a favorable prognosis in comparison to HPV-unrelated OPSCC. Risk classification by Ang is known to predict prognosis of HPV-related OPSCC, but it remains to be studied whether it is useful in predicting prognosis in operatively treated Japanese patients. HPV-related OPSCC treated from 2010 to 2018 in a single Japanese institution by various methods were analyzed. We found that Ang’s risk classification lost its statistical significance in predicting prognosis in the multivariate analysis and it turned out to not be a robust prognostic factor for Japanese patients with HPV-related OPSCC treated by various methods. Additionally, Ang’s risk classification was not prognostically significant in the patients treated by definitive radiation in this series. Smoking appeared to have a larger impact on prognosis in the operatively treated patients than in the radiotherapeutically treated patients.

**Abstract:**

Background: Validity of the risk classification by Ang for human papillomavirus (HPV)-related oropharyngeal squamous cell carcinoma (OPSCC) remains to be studied in the patients treated by modalities other than chemoradiotherapy and in Japanese patients. Materials and Methods: Between 2010 and 2018, 122 patients with HPV-related OPSCC in stages III and IV by the TNM classification 7th edition (TNM-7) were treated curatively at a single institution in Japan. The median age was 62.7 years. Over 50% of the patients underwent surgery with or without adjuvant therapy. The influence of multiple factors on survival was analyzed. Results: The amount of smoking dichotomized at 10 pack-year, which was used in Ang’s risk classification, was not predictive of prognosis, and Ang’s risk classification was not significantly influential on prognosis in multivariate analysis. In the patients treated with definitive radiation therapy, Ang’s risk classification was not predictive of the prognosis in univariate analysis. The impact of smoking was significant only in the patients undergoing the definitive operation. Conclusions: Ang’s risk classification was not robust in predicting the prognosis of general Japanese HPV-related OPSCC patients. The amount of smoking might have different prognostic influences depending on the therapeutic method.

## 1. Introduction

The incidence of human papillomavirus (HPV)-related oropharyngeal squamous cell carcinoma (OPSCC) is increasing rapidly in developed countries [1,2]. In Japan, a similar increase in HPV-related OPSCC has been observed [3]. In contrast to HPV-unrelated OPSCC, HPV-related OPSCC has been reported to have a favorable prognosis [4,5,6,7]. Although smoking is one of the major causes and prognostic factors of HPV-unrelated OPSCC, the prognostic impact of smoking remains controversial in HPV-related OPSCC [8,9,10,11]. A milestone study by Ang et al. revealed the significant impact of smoking expressed in pack-year (PY) on overall survival (OS) and progression-free survival (PFS) in HPV-related OPSCC. According to the risk classification of Ang [4], HPV-related OPSCC is classified into low or intermediate risk for death by the number of PY and N stages by TNM 7th edition (TNM-7). High risk is limited to HPV-unrelated OPSCC. Patients with a smoking history of ≤10 PY and patients of >10 PY but with N0-N2a were classified into the low-risk group. The patients with HPV-related OPSCC with PY >10 and N2b-N3 without distant metastasis were classified into the intermediate-risk group. Ang’s risk classification was derived from selected patients with both HPV-related and -unrelated OPSCC treated exclusively with chemoradiation according to the RTOG 0129 protocol [4]. However, it has not yet been reported whether Ang’s risk classification is also valid in the general HPV-related OPSCC patient population treated with other modalities such as surgery, and a validity analysis has not been performed at all in the general Japanese patient population with HPV-related OPSCC. Robust and reproducible risk classification is important to select the appropriate patients for de-escalation of therapeutic intensity in the HPV-related OPSCC.

In this retrospective study, the robustness and validity of Ang’s risk classification were examined in the patients with HPV-related OPSCC treated with various modalities in a single institution in Japan.

## 2. Materials and Methods

### 2.1. Patients

This single-center retrospective study was approved by the Institutional Review Board (No. 2017-091 and 2018-179). Because of the retrospective nature of this study, informed consent was waived.

Between 2010 and 2018, 240 patients with non-metastatic OPSCC with a known p16 status were treated at a single institution in Japan; of these, 143 (59.6%) had p16-positive OPSCC and 97 (40.4%) had p16-negative OPSCC. To diagnose HPV-related OPSCC, p16 immunohistochemical staining (IHC) was used as a surrogate in this study. An expert head and neck pathologist (YM) diagnosed p16 positivity if nuclear and cytoplasmic staining was apparent in >75% of tumor cells. Moreover, 2 of the 143 patients with HPV-related OPSCC were treated palliatively because of poor performance status and were excluded from this study, and the remaining 141 patients were treated with curative intent. Because the study by Ang et al. analyzed only OPSCC patients in stages more advanced than T1-2N0M0 in TNM-7 [4], 19 patients with T1-T2N0M0 (TNM-7) were excluded from further analysis. The remaining 122 patients with p16-positive OPSCC in stages III and IV in TNM-7 were investigated in this study. The clinical characteristics of these 122 patients are presented in Table 1. More than 80% of patients were men and the median age was 62.7 years (range, 35–83 years). Median hemoglobin before treatment was 13 g/dL for women and 14.5 g/dL for men. Approximately 72% of all patients were current or former smokers. The amount of smoking is expressed in PY with a median of 25.3 PY among smokers and 18.3 PY for all. Synchronous or metachronous secondary cancers were found in 35 patients (28.7%). Alcohol intake and comorbidity were not analyzed in this study because of the lack of detailed information.

The staging was performed with physical examinations, laryngopharyngeal endoscopy, computed tomography (CT), and magnetic resonance imaging (MRI). Positron emission tomography (PET)-CT or PET-MRI was performed in selected patients. More than 70% of the patients had a primary lesion in the tonsil, which was followed by the base of the tongue primary in terms of frequency. Stage classification was performed according to the TNM 8th edition (TNM-8) and N classification (N-7) of the TNM-7. Sixty-seven patients were classified as stage I by the TNM-8, and as stage III and IVA by the TNM-7 (Table 1).

### 2.2. Treatment

Regarding treatment, 65 (53.3%) patients underwent surgery with or without adjuvant radiation therapy (RT) or chemoradiotherapy (Table 1). Primary resection and neck dissection were performed in 62 patients; among them, 7 with the base of the tongue primary underwent bilateral neck dissection. In the remaining three patients, neck lymph node metastasis was managed by neck dissection, while the primary site was irradiated definitively. In 23 of the 65 patients undergoing surgery, postoperative radiation therapy alone was delivered, while 15 patients were treated by postoperative concurrent chemoradiotherapy.

The remaining 57 patients were managed with definitive RT with (47 patients) or without (10 patients) concurrent chemotherapy. Neoadjuvant chemotherapy was administered to 11 of the 47 patients treated with concurrent chemoradiotherapy.

In total, concurrent chemoradiotherapy was performed in 62 patients (definitive 47 + postoperative 15). In 47 patients, tri-weekly cis-platinum (80 mg/m^2^) was administered, while cetuximab was administered to 15 patients with a priming dose of 400 mg/m^2^ and an ensuing weekly dose of 250 mg/m^2^ because of poor renal function. The neoadjuvant chemotherapy regimen was DCF (docetaxel 75 mg/m^2^ on day 1, cis-platinum 75 mg/m^2^ on day1, 5-FU 750 mg/m^2^ from day 1 to day 5, repeated at 21-day intervals) in 9 patients, and CF (cis-platinum 100 mg/m^2^ on day 1 and 5-FU 1000 mg/m^2^ from day 1 to 5, repeated at 21-day intervals) in 2 patients.

Definitive and postoperative RT using 6 MV X-rays from accelerators (Varian, Palo Alto, CA, USA) were performed with intensity-modulated RT in 93 out of 95 patients. In the definitive RT, the gross tumor volume was irradiated with doses between 60 and 70 Gy in conventional fractionation. Prophylactic neck regions were irradiated up to 54 Gy. Postoperative RT to the bilateral necks was delivered in doses between 50 and 60 Gy in a conventional fractionation in cases with positive operative margins, multiple lymph node involvement, and/or extranodal invasion, with a boost dose of 6–10 Gy at the sites of extranodal invasion or positive margins. Doses of <60 Gy were applied in only two patients who were irradiated postoperatively. In total, definitive RT with or without chemotherapy was performed in 57 patients (46.7%) and the remaining 65 patients (53.3%) underwent definitive operation (OP) with or without adjuvant therapy.

### 2.3. Statistical Analysis

The median follow-up period for all patients was 52 months. Progression-free survival (PFS), disease-specific survival (DSS), and overall survival (OS) were calculated using the Kaplan–Meier method, assuming the date of treatment initiation as day 0. For calculation of PFS, recurrence and death from any cause were considered as events. For calculation of DSS, death by tumor was considered as an event, and patients who died without developing recurrence and patients who were alive at the final follow-up were censored. The difference between survival curves was tested by log-rank. In univariate analyses (UVAs), hazard ratio (HR) was calculated in gender, age, hemoglobin, smoking habit, presence of secondary cancer, primary site, T stage (TNM-8), N stages (TNM-7 and TNM-8), clinical stage (TNM-8), treatment, and Ang’s risk classification by using Cox proportional hazard models with PFS, DSS, and OS as endpoints. Dichotomizations with the largest HR were selected from each variable and they further underwent multivariate analyses (MVAs) using Cox proportional hazard regression models. Ang’s risk classification can be considered robust if the risks defined by the classification show statistically significant differences in PFS and OS by both UVA and MVA, since Ang et al. demonstrated the significance of their risk classification in PFS and OS [4]. Additionally, prognostic factors were studied in the patients treated by definitive RT and definitive OP, separately by UVA. Because of the small number of patients in each therapy, MVA was not performed. All statistical analyses were performed using SPSS ver. 26.

## 3. Results

For all 122 patients, PFS, DSS, and OS were 71.9%, 93.0%, and 88.9% at 3 years, and 65.2%, 90.5%, and 85.6% at 5 years, respectively (Figure 1). According to the TNM-8, 3-year PFS, DSS, and OS were 74.5%, 95.2%, and 90.8% for stage I; 76.0%, 91.7%, and 91.7% for stage II; and 62.4%, 89.3%, and 82.4% for stage III, respectively. No significant differences were seen in prognoses between clinical stages according to TNM-8 (Figure 2).

Results of the UVA with HR calculation for each variable are shown in Table 1. Dichotomization points of the variables were chosen to give the largest HRs. The following dichotomization points were selected for further analysis (Table 1): age was divided at 70 years, pretreatment hemoglobin value at the median, smoking habit at 30 PY, T-stage by TNM-8 between T2 and T3, N-stage by TNM-7 between N2a and N2b. Because T-stage and N-stage were analyzed, the clinical stage was not examined further in the ensuing UVA and MVA to prevent confounding. The presence of secondary cancer was categorized as yes or no. The primary site in the oropharynx was classified into the base of the tongue and others. Treatment was divided into definitive RT and definitive OP. Definitive RT included the patients undergoing definitive RT with or without chemotherapy, and definitive OP included the patients treated by surgical resection with or without adjuvant therapies. By the UVA comparing survival curves by log-rank test (Table 2), the amount of smoking with the dichotomization point at 30 PY was statistically significant only in DSS and OS, while the dichotomy at 10 PY, which was used in Ang’s risk classification, had no statistical significance in PFS, DSS, and OS at all (Appendix A). N-stage, dichotomized between N2a and N2b, as used in Ang’s risk classification, was statistically significant in OS (*p* = 0.010) and nearly significant in PFS and DSS with *p*-values of 0.072 and 0.051, respectively (Table 2). Although the smoking amount dichotomized at 10 PY was not statistically significant, Ang’s risk classification reached a statistical significance in PFS (*p* = 0.021) and OS (*p* = 0.016), and almost a statistical significance (*p* = 0.065) in DSS by the UVA. However, the statistical significance of Ang’s risk classification in PFS, DSS, and OS was lost in the MVA (Table 2). Because the DSS and OS of the patients with N0-N2a were 100% (no events), the HRs were not calculated in the MVA, but N-stage seemed to be very significant in DSS and OS. Age dichotomized at 70 years remained statistically significant in DSS and OS in the MVA. It followed that age and N-stage (TNM-7) were the only variables significant in the MVA.

Additionally, the statistical significance of Ang’s risk classification was studied by UVA in the patients treated with definitive RT and definitive OP separately (Table 3). In the patients undergoing definitive OP, PFS, DSS, and OS of the low- and intermediate-risk patients by Ang’s risk classification were different with statistical significance. Three-year PFS, DSS, and OS were 77.8%, 100%, and 100% in the low-risk patients, and 51.7% (*p* = 0.017), 81.2% (*p* = 0.010), and 75.4% (*p* = 0.003) in the intermediate-risk patients, respectively (Figure 3). In the patients with definitive OP, DSS and OS showed statistically significant differences by the smoking amount with dichotomization of 30 PY. In contrast, surprisingly, the prognoses were not at all impacted by Ang’s risk classification in the patients treated by definitive RT, and smoking had no statistical influence on prognoses (Figure 3).

## 4. Discussion

With this study, we aimed to verify the validity of Ang’s risk classification in patients with HPV-related OPSCC treated with various modalities in a single institution in Japan.

HPV-related OPSCC is known to have different etiologies and therapeutic responses than HPV-unrelated OPSCC. HPV-related OPSCC is caused by infection with high-risk HPV and responds well to therapeutic interventions with a favorable prognosis, in contrast to HPV-unrelated OPSCC [4,5,6,7]. The TNM-8 separates HPV-related and HPV-unrelated OPSCC, taking into account the etiological and prognostic differences. The TNM-8 uses p16 positivity as a surrogate marker for HPV infection, similar to the methods employed in this study. Although some studies have reported better differentiation of OS by the stages of TNM-8 than by the stages of TNM-7 [12,13,14,15], other reports indicated that prognosis is inadequately differentiated by TNM-8 [16,17]. The current study showed that the stages defined by TNM-8 were not able to differentiate PFS, OS, and DSS. The OS of patients with stage III in this study was better than that of the other reported series [12,13,14,15]. DeFelice et al. also demonstrated a favorable 5-year OS of 86.2% in patients with stage III HPV-related OPSCC and denied the utility of TNM-8 [17].

Because of the favorable prognosis of HPV-related OPSCC, therapeutic de-escalation has been attempted [18,19,20]. However, to demonstrate the feasibility of de-escalated therapeutic strategies, it is imperative to select patients who can be managed with less intensive therapy [18,20]. To identify appropriate patients, proper prognostic criteria are indispensable, and the TNM-8 staging seemed to be inadequate to extract HPV-related OPSCC patients with good prognosis who would be candidates for de-escalated treatment.

The risk classification proposed by Ang et al. was derived from selected patients treated with chemoradiation therapy in a randomized clinical trial [4]. The smoking amount and N-stage, which are used as criteria in Ang’s risk classification, were statistically significant prognosticators in PFS and OS in both UVA and MVA in their study [4]. Several studies have demonstrated the validity of Ang’s risk classification in HPV-related OPSCC patients managed by radiation therapy with and without chemotherapy [17,21,22,23,24], but its validity has not been confirmed in surgically treated patients and in the general Japanese patient population. Ang’s risk classification is defined by the amount of smoking less than or equal to 10 PY or more than 10 PY, and N-stage of N0-N2a or N2b-N3 according to TNM-7. Although the amount of smoking dichotomized at 10 PY did not influence PFS, DSS, and OS, and N-stage dichotomization between N2a and N2b brought about statistical significance only in OS by UVA in this study (Table 2), the low- and intermediate-risk patients by Ang’s risk classification showed statistically significant differences in PFS and OS in the UVA. However, the statistical significance of Ang’s risk classification was lost in the MVA. In the MVA, N-stage dichotomized between N2a and N2b and age dichotomized at 70 years seemed to affect DSS and OS, but the smoking amount was not statistically significant. While the smoking amount, N-stage by TNM-7, and Ang’s risk classification proved to be predictive of various combinations of outcomes within the UVA and MVA analyses, none met the criteria of consistently predicting each PFS and OS across UVA and MVA analysis. Thus, Ang’s risk classification does not meet our definition of robustness and cannot necessarily be generalized and considered robust to all HPV-related OPSCC patients who receive a variety of treatment modalities. Instead, Ang’s risk classification appears to be appropriate when certain patient characteristics and treatment conditions are met.

In contrast, in the patients treated by the definitive OP, the low- and intermediate-risk patients by Ang’s risk classification showed statistically significant differences in PFS, DSS, and OS in the UVA, although a low number of the patients precluded MVA. However, even in the patients treated by definitive OP, the amount of smoking and N-stage dichotomy, both of which are criteria of Ang’s risk classification, were not unanimously significant in PFS, DSS, and OS. Because of lack of MVA, the robustness of Ang’s risk classification in the patients treated by the definitive OP cannot be discussed. However, Ang’s risk classification appears to be useful in this patient cohort. Surprisingly, in the patients undergoing the definitive RT in this study, no significant prognostic variables were found including Ang’s risk classification. In the patients treated by the definitive RT, the amount of smoking had no impact on prognoses, while DSS and OS seemed to be influenced by the amount of smoking in the patients undergoing the definitive OP. It seems that smoking has a different impact depending on the therapeutic modality. The definitive RT might overcome the predicaments posed by smoking. However, there are reports which denied the different effects of smoking on the different therapeutic modalities [25,26,27], and Kimura et al. suggested conversely that surgical treatment was influenced less by smoking in comparison to radiation therapy in their series [28]. It should be further studied whether the prognostic influence of smoking differs depending on therapeutic modalities.

There are two points that must be mentioned in this Japanese patient series. First, almost all Japanese series of HPV-related OPSCC reported median age of patients older than 60 years [3,7,29,30] in contrast to Western series where the median age was always in the 50s. It remains to be studied whether Japanese people are infected by HPV at a later age than Western people or if the latent period of HPV oncogenesis is longer in Japanese people. As was shown in this study, the prognostic significance of age in HPV-related OPSCC was repeatedly demonstrated in many studies [4,31,32,33,34,35,36]. Many Western studies use age dichotomy at 50 years [4,32], although this study showed statistical significance in age division at 70 years. Huang (Canadian patients) [36] and Lassen (Canadian and Danish patients) [35] used the same age division at 70 years. It remains to be studied how the median age influences the prognostic significance of age in HPV-related OPSCC.

Second, in this study, the smoking amount was dichotomized at 30 PY, which showed the largest HR in UVA, but its prognostic significance was lost in MVA. The smoking habits of the United States, Europe, and Japan are different [37]. Many studies demonstrated the prognostic significance of smoking amount dichotomized at 10 PY [4,21,31], but the dichotomization point seems to be proportional to the median PY. If the median PY is larger, then the dichotomization point is also in higher PY. For example, Gronhoji et al. used dichotomization at 30 PY and the median PY was 30 in their series [38]. Ang et al. used 10 PY with a median PY of 5.25 in their series [4]. In this study, 30 PY was used with the median PY of 18.3. The prognostic impact of smoking must be further studied in HPV-related OPSCC in correlation to the general smoking habit of the population.

The retrospective nature of this study limits its value. Furthermore, this study dealt with the patients treated in a single Japanese institution dedicated specifically to cancer therapy. It remains to be seen whether this trend holds across institutions with their own oncological teams treating Japanese OPSCC patients using different surgical approaches and chemoradiotherapy regimens. This study used p16 positivity as a surrogate for HPV infection, and HPV-DNA and m-RNA analyses were not performed. There are numerous reports disclosing discrepancies between p16 positivity and HPV-DNA detection [31,39]. p16-positive but HPV-DNA-negative patients show poorer prognosis in comparison to p16-positive and HPV-DNA-positive patients. HPV-DNA analysis of the same patient cohort is planned and will be published in the near future. Additionally, important information regarding alcohol consumption [17,40] and comorbidities [31] is lacking in this study. Both were reported to be significant prognosticators, and without this information, the value of this study might be limited.

## 5. Conclusions

In the current study, TNM-8 could not adequately differentiate the prognosis of patients with HPV-related OPSCC in Japan. Moreover, Ang’s risk classification turned out not to be a robust prognostic factor in the general Japanese patients with HPV-related OPSCC treated by various modalities, including surgery. Additionally, Ang’s risk classification was not at all prognostic in the patients treated by the definitive RT, while Ang’s risk classification appears to be useful to predict prognoses in the patients undergoing the definitive OP. The effect of smoking might be different depending on the therapeutic modalities. To select the optimal patients for de-escalated treatment, Ang’s risk classification is not simply generalizable in Japanese patients.

## Figures and Tables

**Figure 1 cancers-14-02442-f001:**
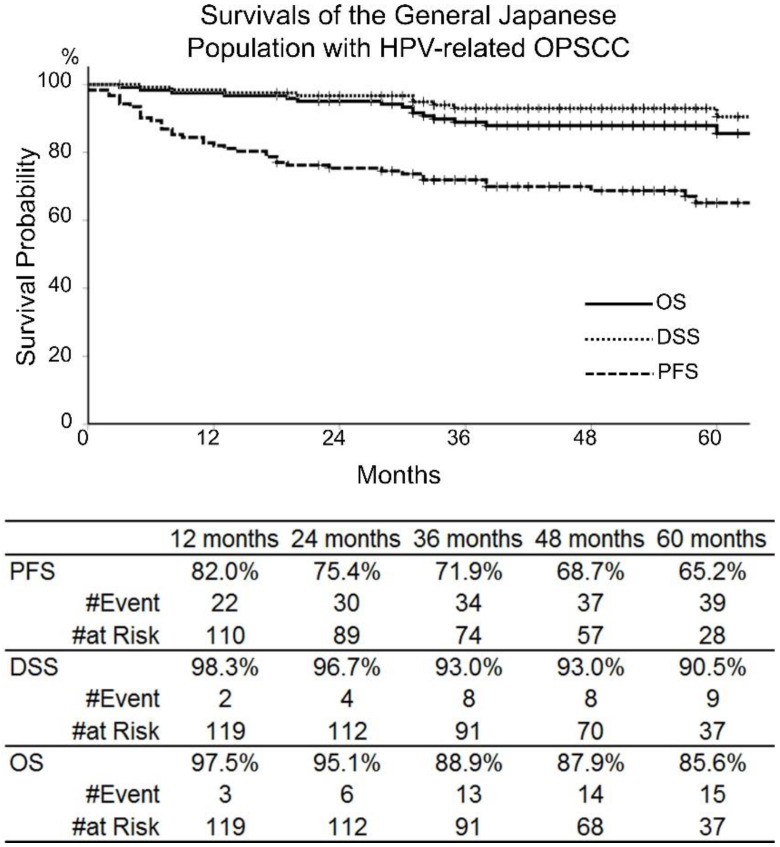
Progression-free survival (PFS), disease-specific survival (DSS), and overall survival (OS) of the 122 patients with human papillomavirus-related oropharyngeal squamous cell carcinoma (HPV-related OPSCC) in the clinical stages III and IV, according to the TNM 7th edition, who were treated between 2010 and 2018.

**Figure 2 cancers-14-02442-f002:**
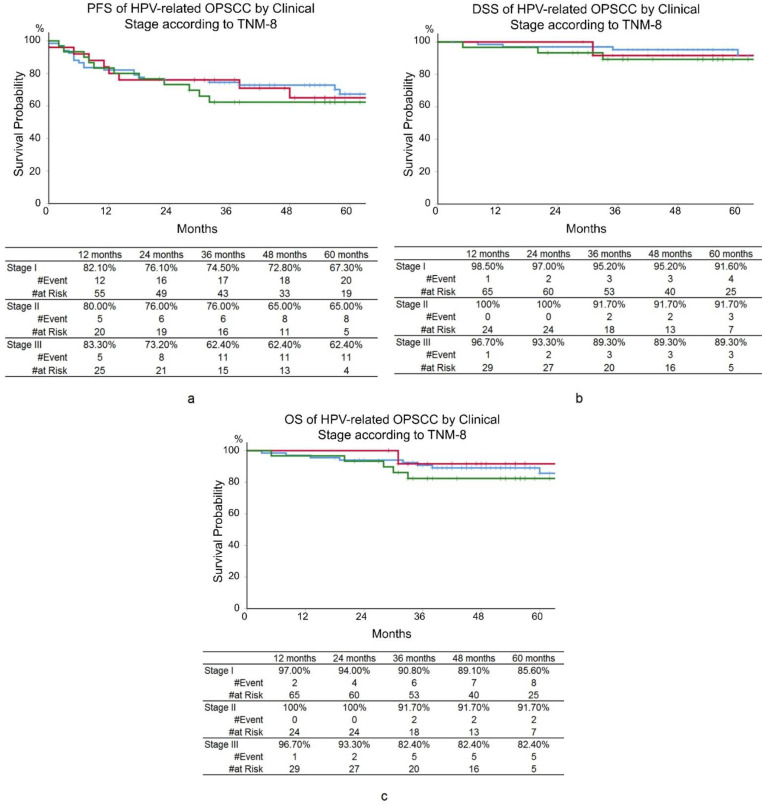
Kaplan–Meier curves. (**a**) Progression-free survival (PFS). (**b**) Disease-specific survival (DSS). (**c**) Overall survival (OS) of human papillomavirus-related oropharyngeal squamous cell carcinoma (HPV-related OPSCC) according to the clinical stage by TNM 8th edition. Blue lines: stage I, red lines: stage II, green lines: stage III. No significant differences were seen in prognoses between clinical stages according to TNM-8.

**Figure 3 cancers-14-02442-f003:**
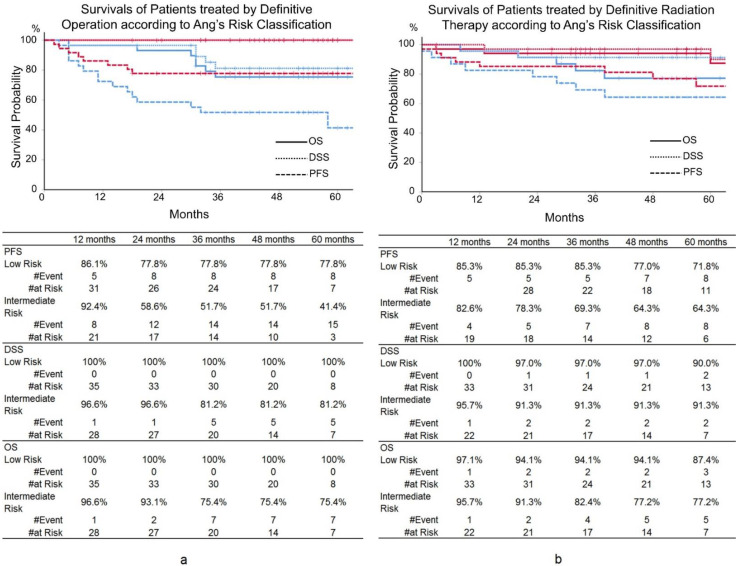
Progression-free survival (PFS), disease-specific survival (DSS), and overall survival (OS) in the patients treated by operation and radiation according to Ang’s risk classification. (**a**) The patients treated by operation. (**b**) The patients treated by definitive radiation. Red line: low risk, blue line: intermediate risk.

**Table 1 cancers-14-02442-t001:** Patient characteristics and univariate analysis with hazard ratios of the possible prognostic factors. If not indicated, the first covariate was reference (HR = 1).

			PFS		DSS		OS	
		n (%)	HR (95% CI)	*p*	HR (95% CI)	*p*	HR (95% CI)	*p*
Gender								
	male/female ^&^	99 (81.1%)/23 (18.9%)	0.944 (0.629–1.417)	0.781	1.075 (0.228–5.063)	0.928	0.931 (0.267–3.241)	0.911
Age								
	median	62.7 years	1.020 (0.991–1.050) **	0.175	1.073 (1.007–1.144) **	0.031 *	1.065(1.015–1.118) **	0.01 *
	<50 years/≥50 years	16 (13.1%)/106 (86.9%)	1.091 (0.682–1.742)	0.718	5.051 (0.106–250)	0.411	1.647 (0.599–4.525)	0.334
	<60 years/≥60 years	51 (41.8%)/71 (58.2%)	1.161 (0.842–1.600)	0.363	1.304 (0.663–2.564)	0.442	1.543(0.881–2.705)	0.129
	<70 years/≥70 years ^&^	92 (75.4%)/30 (24.6%)	1.277 (0.931–1.751)	0.129	2.169 (1.151–4.082)	0.017 *	1.873(1.162–3.017)	0.01 *
Hemoglobin								
	female median	13 g/dL						
	male median	14..5 g/dL						
	>median/≤median ^&^	63 (51.6%)/59 (48.4%)	1.072 (0.791–1.454)	0.654	1.191 (0.632–2.245)	0.589	1.325 (0.805–2.181)	0.268
Smoking habit								
	ever/never	88 (72.1%)/34 (27.9%)	1.072 (0.765–1.501)	0.687	1.064 (0.541–2.093)	0.857	1.075 (0.637–1.815)	0.787
	median PY for ever smoker	25.3 PY						
	median PY for all	18.3 PY	1.005 (0.996–1.014) **	0.270	1.010 (0.993–1.027) **	0.256	1.012 (1.000–1.025) **	0.043 *
	≤10 PY/>10 PY	48 (39.3%)/74 (60.7%)	1.156 (0.839–1.594)	0.376	1.244 (0.632–2.447)	0.528	1.238 (0.734–2.087)	0.423
	≤20 PY/>20 PY	63 (51.6%)/59 (40.4%)	1.155 (0.852–1.566)	0.352	1.588 (0.807–3.125)	0.18	1.388 (0.843–2.284)	0.198
	≤30 PY/>30 PY ^&^	84 (68.9%)/38 (31.1%)	1.208 (0.884–1.650)	0.235	1.865 (0.990–3.514)	0.054	1.801 (1.110–2.923)	0.017 *
Secondary cancer								
	no/yes ^&^	87 (71.3%)/35 (28.7%)	1.037(0.751–1.433)	0.826	1.579 (0.850–2.941)	0.148	1.876 (1.157–3.040)	0.011 *
Primary site								
	base of tongue/other ^&^	28 (23.0%)/94 (77.0%)	1.133 (0.770–1.667)	0.525	1.066 (0.491–2.316)	0.871	1.464 (0.700–3.063)	0.312
T-stage (TNM-8)								
	T1	17 (13.9%)	0.681 (0.209–2.216)	0.524	0.760 (0.069–8.402)	0.823	0.377 (0.042–3.375)	0.383
	T2	61 (50.0%)	1.092 (0.501–2.378)	0.825	0.771 (0.140–4.251)	0.766	0.891 (0.273–2.909)	0.848
	T3	16 (13.1%)	1.428 (0.532–3.838)	0.825	2.389 (0.399–14.319)	0.340	1.225 (0.274–5.475)	0.791
	T4	28 (23.0%)	1		1		1	
	T1-2/T3-4 ^&^	78 (63.9%)/44 (36.1%)	1.074(0.785–1.468)	0.656	1.414 (0.758–2.639)	0.276	1.178 (0.725–1.912)	0.510
	T1-3/T4	94 (77.0%)/28 (23.0%)	0.967 (0.668–1.401)	0.861	0.981 (0.451–2.137)	0.963	1.081 (0.6161–1.898)	0.786
N-stage								
	N0 (TNM-8)	5 (4.1%)	0.311 (0.051–1.883)	0.204	0	0.971 ^?^	0	0.972 ^?^
	N1 (TNM-8)	95 (77.8%)	0.237 (0.072–0.786)	0.019*	6.851 (0–)	0.980	11.200 (0–)	0.976
	N2 (TNM-8)	18 (14.8%)	0.227 (0.056–0.917)	0.037 *	17.452 (0–)	0.970	18.853 (0–)	0.971
	N3 (TNM-8)	4 (3.3%)	1		1		1	
	N0-2a/N2b-3 (TNM-7) ^&^	33 (27.0%)/89 (73.0%)	1.437 (0.9578–2.16)	0.080	33.333 (0.094–)	0.242 ^?^	5.813 (0.620–55.55)	0.123 ^?^
	N0-2b/N2c-3 (TNM-7)	100 (82.0%)/22 (18.0%)	1.126 (0.7782–1.629)	0.529	1.730 (0.917–3.257)	0.090	1.362 (0.8071–2.299)	0.248
Clinical stage (TNM-8)								
	I	67 (54.9%)	1		1		1	
	II	25 (20.5%)	0.947 (0.374–2.396)	0.873	2.057 (0.360–11.958)	0.266	0.9149 (0.207–4.046)	0.764
	III	30 (24.6%)	1.034 (0.687–1.555)	0.740	1.618 (0.693–3.77)	0.589	1.105 (0.578–2.114)	0.659
Treatment								
	OP alone	27 (22.1%)	1.679 (0.464–6.074)	0.430	0.430 (0.059–3.113)	0.404	0.418 (0.058–2.997)	0.385
	OP + RT or OP + CRT	38 (31.2%)	1.420 (0.405–4.974)	0.583	0.419 (0.070–2.509)	0.341	0.700 (0.136–3.613)	0.670
	RT alone	10 (8.2%)	1		1		1	
	CRT	36 (29.5%)	1.020 (0.284–3.666)	0.976	0.422 (0.071–2.531)	0.345	1.131 (0.240–5.330)	0.876
	NeoCT + CRT	11 (9.0%)	1.126 (0.226–5.618)	0.885	0	0.985 ^?^	0	0.981 ^?^
	definitive RT/definitive OP *** ^&^	57 (46.7%)/65 (53.3%)	1.214 (0.889–1.656)	0.223	0.976 (0.523–1.821)	0.938	0.813 (0.500–1.323)	0.405
Ang’s risk classification								
	low/intermediate ^&^	70 (57.4%)/52 (42.6%)	2.021 (1.095–3.729)	0.024 *	3.312 (0.856–12.813)	0.083	3.348 (1.179–9.508)	0.023 *

* *p* < 0.05, ** treated as a continuous variable, *** definitive RT includes RT alone, CRT, and NeoCT-CRT, and definitive OP includes OP alone, OP + RT and OP + CRT. ? calculation result was not relibale because survival rate was 100%, & selected variables for further analysis. CI: confidence interval, CRT: chemoradiation therapy, DSS: disease-specific survival, HR: hazard ratio, NeoCT: neoadjuvant chemotherapy, OP: operation, OS: overall survival, PFS: progression-free survival, PY: pack-year, RT: radiation therapy.

**Table 2 cancers-14-02442-t002:** Univariate and multivariate analysis using the selected variables with PFS, DSS, and OS as endpoints.

		PFS				DSS				OS			
		UVA		MVA		UVA		MVA		UVA		MVA	
		3-Year PFS	*p*	HR (95% CI)	*p*	3-Year DSS	*p*	HR (95% CI)	*p*	3-Year OS	*p*	HR (95% CI)	*p*
Gender													
	female	73.9%	0.780	1	0.565	92.4%	0.927	1	0.379	88.3%	0.911	1	0.649
	male	71.4%		1.770 (0.316–1.876)		95.5%		0.407 (0.055–3.012)		91.3%		0.700 (0.150–3.257)	
Age													
	<70 years	74.8%	0.124	1	0.118	96.6%	0.008 *	1	0.009 *	93.2%	0.006 *	1	0.010 *
	≥70 Yeas	63.3%		1.681 (0.876–3.226)		81.9%		7.246 (1.631–32.258)		76.2%		3.984 (1.399–11.364)	
Hemoglobin													
	>median	72.9%	0.652	1	0.511	93.2%	0.589	1	0.318	86.6%	0.262	1	0.257
	≤median	70.9%		1.261 (0.632–2.516)		92.8%		2.270 (0.454–11.358)		91.2%		2.003 (0.603–6.659)	
Smoking habit													
	≤30 PY	76.0%	0.230	1	0.385	97.5%	0.040 *	1	0.099	95.1%	0.012 *	1	0.112
	>30 PY	62.8%		1.403 (0.654–3.011)		82.7%		6.997 (0.696–70.376)		75.4%		3.745 (0.736–19.048)	
Secondary cancer													
	no	72.3%	0.825	1	0.556	95.3%	0.134	1	0.567	94.2%	0.006 *	1	0.158
	yes	71.4%		0.809 (0.399–1.639)		87.2%		1.471 (0.393–5.495)		76.9%		2.141 (0.743–6.173)	
Primary site													
	base of tongue	78.6%	0.523	1	0.739	92.6%	0.871	1	0.952	92.6%	0.300	1	0.444
	others	70.0%		1.148 (0.508–2.594)		93.1%		0.948 (0.165–5.436)		87.9%		1.865 (0.378–9.216)	
T-stage (TNM-8)													
	T1-T2	74.3%	0.655	1	0.512	94.6%	0.266	1	0.146	90.8%	0.508	1	0.412
	T3-T4	67.7%		1.259 (0.632–2.513)		90.1%		3.012 (0.682–13.333)		85.7%		1.597 (0.522–4.878)	
N-stage (TNM-7)													
	N0-N2a	84.8%	0.072	1	0.135	100.0%	0.051	1	0.959 ^?^	100.0%	0.010 *	1	0.951 ^?^
	N2b-N3	67.2%		2.252 (0.777–6.536)		90.3%		0		84.8%		0	
Treatment													
	Definitive RT **	78.7%	0.218	1	0.075	94.6%	0.938	1	0.230	89.3%	0.402	1	0.678
	Definitive OP **	66.0%		1.925 (0.937–3.955)		91.6%		2.984 (0.500–17.818)		88.6%		1.295 (0.382–4.386)	
Ang’s risk classification													
	low	81.40%	0.021 *	1	0.575	98.60%	0.065	1	0.407	97.10%	0.016 *	1	0.478
	intermediate	59.34%		1.298 (0.522–3.229)		85.60%		0.300 (0.017–5.177)		78.40%		0.499 (0.073–3.398)	

* *p* < 0.05, ** definitive RT includes RT alone, CRT, and NeoCT-CRT, and definitive OP includes OP alone, OP + RT and OP + CRT. ? calculation result was not reliable because survival rate was 100%; CI: confidence interval, CRT: chemoradiation therapy, DSS: disease-specific survival, HR: hazard ratio, MVA: multivariate analysis, NeoCT: neoadjuvant chemotherapy, OP: operation, OS: overall survival, PFS: progression-free survival, PY: pack-year, RT: radiation therapy, UVA: univariate analysis.

**Table 3 cancers-14-02442-t003:** Separate univariate analyses for the patients undergoing definitive surgery and radiation therapy.

		Definitive OP **							Definitive RT **						
		n	3-Year PFS	*p*	3-Year DSS	*p*	3-Year OS	*p*	n	3-year PFS	*p*	3-year DSS	*p*	3-year OS	*p*
Gender															
	female	8	62.5%	0.800	100%	0.366	100%	0.291	15	80.0%	0.840	92.9%	0.522	86.7%	0.780
	male	57	66.4%		90.2%		86.9%		42	78.2%		95.2%		90.2%	
Age															
	<70 years	52	71.2%	0.073	96.0%	0.100	96.0%	<0.001 *	40	79.5%	0.730	97.4%	0.163	89.6%	0.542
	≥70 years	13	46.2%		70.1%		59.3%		17	76.5%		88.2%		88.2%	
Hgb															
	>median	37	64.9%	0.881	91.1%	0.946	88.7%	0.941	22	86.4%	0.369	95.5%	0.506	95.5%	0.265
	<median	28	67.5%		92.2%		88.5%		35	73.7%		94.1%		85.2%	
Smoking habit															
	≤30 PY	45	71.0%	0.062	97.6%	0.011 *	95.3%	0.012 *	39	82.1%	0.912	97.4%	0.829	94.9%	0.332
	>30 PY	20	54.5%		77.5%		73.2%		18	72.2%		88.9%		71.8%	
	≤10 PY	24	75.0%	0.242	100%	0.078	100%	0.038 *	24	83.3%	0.944	95.7%	0.390	91.7%	0.500
	>10 PY	41	60.7%		86.6%		82.2%		33	75.6%		93.9%		87.8%	
Secondary cancer															
	no	48	68.6%	0.730	95.7%	0.077	95.7%	0.006 *	39	76.8%	0.988	94.7%	0.702	92.3%	0.239
	yes	17	58.8%		80.0%		70.6%		18	83.3%		94.4%		83.3%	
Primary site															
	base of tongue	14	64.3%	0.722	85.1%	0.286	85.1%	0.590	14	92.9%	0.185	100%	0.252	100%	0.088
	others	51	66.6%		93.4%		89.7%		43	74.1%		92.9%		85.7%	
T-stage															
	T1-2	49	71.4%	0.214	95.7%	0.044 *	93.6%	0.035 *	29	79.2%	0.957	92.9%	0.756	86.1%	0.288
	T3-4	16	50.0%		79.4%		74.5%		28	78.3%		96.4%		92.6%	
N-stage															
	N0-N2a	21	85.7%	0.031 *	100%	0.104	100%	0.058	12	83.3%	0.680	100%	0.203	100%	0.073
	N2b-N3	44	56.6%		87.3%		83.2%		45	77.6%		93.2%		86.4%	
Ang’s risk classification															
	low	36	77.8%	0.017 *	100%	0.010 *	100%	0.003 *	34	85.3%	0.500	97.0%	0.915	94.1%	0.467
	intermediate	29	51.7%		81.2%		75.4%		23	69.3%		91.3%		82.4%	

* *p* < 0.05, ** definitive RT includes RT alone, CRT, and NeoCT-CRT, and definitive OP includes OP alone, OP + RT and OP + CRT. CRT: Chemoradiation therapy, DSS: disease-specific survival, HR: hazard ratio, NeoCT: neoadjuvant chemotherapy, OP: operation, OS: overall survival, PFS: progression-free survival, PY: pack-year, RT: radiation therapy, UVA: univariate analysis.

## Data Availability

The datasets used and analyzed during the current study are available from the corresponding author on reasonable request after the approval of the National Cancer Center review board.

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
