# Peer review of "Non-Robustness of Ang’s Risk Classification in Human Papillomavirus-Related Oropharyngeal Squamous Cell Carcinoma in Japanese Patients"

_cancers, 2022, doi:10.3390/cancers14102442_

Round 1

Reviewer 1 Report

The submitted manuscript is a retrospective study of 122 HPV-associated OPSCC patients treated at a Japanese institution between 2010 and 2018. The authors analyzed several patient characteristics and various treatment modalities to identify variables that were predictive of OS, PFS, and DSS in univariate (UV) and multivariate (MV) analyses. These analyses were also performed on subgroups of these patients that were treated either with definitive surgery or definitive radiotherapy. Furthermore, the authors tested whether Ang’s risk, a proposed risk classification for OPSCC patients first published in 2010, is robust enough to be applied generally to Japanese OPSCC patients.

The proposed investigation has merit, as predictive variables need to be rigorously tested before they are implemented clinically and used to guide treatment. However, there are significant aspects of this manuscript that need to be addressed before recommending for publication.

Line-specific comments

On Line 62, please explicitly define “The remaining patients with HPV-related OPSCC without distant metastasis” (those with PY > 10 and N2b-N3?)

On Line 63, authors state “Ang’s risk classification was derived from highly selected patients…”. However, patient recruitment for RTOG-0129, the trial that Ang’s risk was derived from, is for a randomized, multicenter phase III clinical trial. There were provided exclusion criteria, but none unusual enough to classify patient recruitment as “highly selective” relative to the standard clinical trial.

On Line 85, the authors state “The remaining 122 patients with p16-positive OPSCC in stages III and IV in TNM-7 were investigated in this study” but on Line 96, the authors state “Stage classification was performed according to the TNM 8th edition (TNM-8) and N classification (N-7) of the TNM-7. Sixty-seven patients were classified as stage I by the TNM-8 (Table 1)”. Please clarify which staging was used (TNM-7 or TNM-8) and if the 67 patients that were classified as Stage I by TNM-8 were classified as Stage II or III by TNM-7. Were these patients included in the study?

What does “reference” mean in Table 1?

For the paragraph beginning on Line 112, move “In 47 patients, triweekly cis-platinum (80 mg/m2) was administered” to Line 107 before talking about neoadjuvant chemotherapy (assuming these are referring to the same 47 patients in this section). Also move “cetuximab was administered to 15 patients with a priming dose of 400 mg/m2 and an ensuing weekly dose of 250 mg/m2 because of poor renal function” to the end of Line 105 (assuming these are referring to the same 15 patients in this section).

On Line 147, please revise “There could not be seen any statistically significant differences in prognoses between clinical stages according to TNM-8” to “No significant differences were seen in prognoses between clinical stages according to TNM-8”.

On Line 179: “while the dichotomy at 10 PY, which was used in Ang’s risk classification, had no statistical significance in PFS, DSS, and OS at all (data not shown).” I think this data needs to be included to compel the reader.

On Line 181: “N-stage, dichotomized between N2a and N2b, as used in Ang’s risk classification, was statistically significant in OS and nearly significant in PFS and DSS with p-values of 0.072 and 0.051, respectively” please report the p-value for OS.

On Line 298, the author states that “From the UVA and MVA, it appears quite difficult in this study to derive the risk classification with a combination of smoking amount and N-stage. It follows that Ang’s risk classification is not a reliable and robust prognostic factor in real-world HPV-related OPSCC of Japanese patients.” This is an exaggerated interpretation of the data. In the author’s results, N-stage and Ang’s risk did quite well in predicting outcomes in UVA, and N-stage did well in MVA. The authors set a strict criteria for “robust”, which was defined as “Ang’s risk classification can be considered robust, if the classification itself shows statistical significance both in UVA and MVA, and the 2 variables in the classification criteria (smoking amount and N-7) both are statistically significant in predicting prognosis as well.” There is also no mention of the author’s definition of “reliable”. I think a fairer assessment of the data is as follows. “While the smoking amount, N-stage, and Ang’s risk proved to be predictive of various combinations of outcomes within the UVA and MVA analyses, none met the criteria of consistently predicting each of PFS, DSS, and OS across UVA and MVA analyses. Thus, Ang’s risk does not meet our definition of robust and cannot necessarily be generalized and considered robust to all HPV-associated OPSCC patients who receive a variety of treatment modalities. Instead, Ang’s risk appears to be appropriate when certain patient characteristics and treatment conditions are met.”  

On Line 304, the author states “However, even in the patients treated by definitive OP, the amount of smoking and N-stage dichotomy, both of which are criteria of Ang’s risk classification, were not unanimously significant in PFS, DSS, and OS. Therefore, Ang’s risk classification appears to be not so robust even in the patients treated by the definitive OP.” This sentence should be significantly altered. Just a few lines before, the author states “patients treated by the definitive OP, the low and intermediate risk patients by Ang’s risk classification showed statistically significant differences in PFS, DSS, and OS in the UVA, although a low number of patients precluded MVA.” Therefore, the author cannot claim that the Ang’s risk is not robust in the definitive OP group. Perhaps if there was enough power to run MVA, which then demonstrated Ang’s risk was not predictive of survival, the authors could argue the point. But as the data stands, it is not appropriate to do so. Lastly, I don’t think testing for “robustness” here is appropriate. Prior to this subgroup analysis, robustness referred to the entire patient cohort regardless of treatment modality. The authors should stick with this definition and use a different adjective to describe Ang’s risk in the scope of the definitive OP and definitive RT groups.

General comments

Figures and Tables need to be cleaned up and look more professional. For example, remove grid lines from survival curves, increase font to be more legible, etc.

Please change all instances of “real-world patients” to “the general HPV-related OPSCC patient population” or something analogous.

Please specify any potential sources of bias. For example, this was a single-institution study. It remains to be seen whether this trend holds across institutions with different surgical teams that treat Japanese OPSCC patients or different chemoradiation regimens.

In general, I believe the definition of “robust” needs to be altered. If the authors only mention the robustness of Ang’s risk in the title of the paper, then the definition of “robust” should only include the predictive power of Ang’s risk in UVA and MVA. I don’t think it is correct to include smoking amount and N-stage as additional criteria for “robust”- even if these variables go into the calculation of Ang’s risk. It is fine to note the discrepancy when Ang’s risk is predictive when smoking amount and N-stage are not. But to state that Ang’s risk is not robust if it does in fact predict outcomes in UVA and MVA doesn’t seem appropriate. Also, since Ang’s risk only considered PFS and OS (DSS was a secondary endpoint in the general RTOG-0129 trial), I don’t think DSS should be included as a criteria when trying to determine its validity.

Author Response

The authors deeply appreciate Reviewer 1 for the extensive in-detailed and insightful remarks and comments. We modified the manuscript according to the suggestions of Reviewer 1 as follows.

  • Reviewer 1: On line 62, please explicitly define “The remaining patients with HPV-related OPSCC without distant metastasis” (those with PY >10 and N2b-N3?)

Response of the authors: According to reviewer’s suggestion, line 62 was changed to “The patients with HPV-related OPSCC with PY > 10 and N2b-N3 without distant metastases were classified into the intermediate risk group.”

  • Reviewer 1: On Line 63, authors state “Ang’s risk classification was derived from highly selected patients…”. However, patient recruitment for RTOG-0129, the trial that Ang’s risk was derived from, is for a randomized, multicenter phase III clinical trial. There were provided exclusion criteria, but none unusual enough to classify patient recruitment as “highly selective” relative to the standard clinical trial.

Response of the authors: According to reviewer’s suggestion, “highly selected” was changed to “selected”, because RTOG 0129 excluded the patients with kidney dysfunction but in this study such patients were included and underwent C-mab or CBDCA instead of CDDP. Therefore the term “selected” was left untouched. In Line 283, also “highly selected” was changed to “selected”.

  • Reviewer 1: On Line 85, the authors state “The remaining 122 patients with p16-positive OPSCC in stages III and IV in TNM-7 were investigated in this study” but on Line 96, the authors state “Stage classification was performed according to the TNM 8th edition (TNM-8) and N classification (N-7) of the TNM-7. Sixty-seven patients were classified as stage I by the TNM-8 (Table 1)”. Please clarify which staging was used (TNM-7 or TNM-8) and if the 67 patients that were classified as Stage I by TNM-8 were classified as Stage II or III by TNM-7. Were these patients included in the study?

Response of the authors: I am very sorry for the very confusing description. As written in line 83-, to comply to the RTOG 0129, patients in T1-2N0 which corresponds to Stage I and II in TNM-7 were excluded. Therefore in this study, only the patients in stages III and IV by TNM-7 were studied. However, in this study clinical stage of TNM-8 was used to classify, so even stage I by TNM-8 would be classified into stage III and IVa by TNM-7. To make it clear, in line 98-99 following sentence was added. “Sixty-seven patients were classified as stage I by the TNM-8, while they belonged to Stage III and IVa by the TNM-7.”  

  • Reviewer 1: What does “reference” mean in Table 1?

Response of the authors: Thank you for your question. “reference” was used in Table 1 and 2. “Reference” meant covariate supposed to have hazard ratio (HR) supposed to be 1, and HRs for other covariates were calculated in comparison to the reference HR of 1. As for clarification, Table 1 legend was changed to “Patient characteristics and univariate analysis with hazard ratios of the possible prognostic factors. If not indicated, the first covariate was reference (HR = 1).” Furthermore, “references” in the content of Tables 1 and 2 were changed to “1”.

  • Reviewer 1: For the paragraph beginning on Line 112, move “In 47 patients, triweekly cis-platinum (80 mg/m2) was administered” to Line 107 before talking about neoadjuvant chemotherapy (assuming these are referring to the same 47 patients in this section). Also move “cetuximab was administered to 15 patients with a priming ose of 400 mg/m2 and an ensuing weekly dose of 250 mg/m2 because of poor renal function” to the end of Line 105 (assuming these are referring to the same 15 patients in this section).

Response of the authors: We thank you for your valuable suggestions and are sorry for the complicated and confusing description. According to the suggestions of the reviewer, sentences from 106 were changed as follows to clarify the number of patients treated by each modality, “In 23 of the 65 patients undergoing surgery, postoperative radiation therapy alone was delivered, while 15 patients were treated by postoperative concurrent chemoradiotherapy.

The remaining 57 patients were managed with definitive RT with (47 patients) or without (10 patients) concurrent chemotherapy. Neoadjuvant chemotherapy was administered to 11 of the 47 patients treated with concurrent chemoradiotherapy.

In total, concurrent chemoradiotherapy was performed in 62 patients (definitive 47 + posoperative 15). In 47 patients, triweekly cis-platinum (80 mg/m2) was administered, while cetuximab was administered to 15 patients with a priming dose of 400 mg/m2 and an ensuing weekly dose of 250 mg/m2 because of poor renal function.” As suggested by the reviewer, the description of neoadjuvant chemotherapy follows after the explanation of concurrent chemoradiation therapy. 

  • Reviewer 1: On Line 147, please revise “There could not be seen any statistically significant differences in prognoses between clinical stages according to TNM-8” to “No significant differences were seen in prognoses between clinical stages according to TNM-8”.

Response of the authors: It is deeply grateful of Reviewer 1 for an educative suggestion. The authors modify the sentence as follows, “No significant differences were seen in prognoses between clinical stages according to TNM-8” line 157(because of the necessary changes, line number is different).

  • Reviewer 1: On Line 179: “while the dichotomy at 10 PY, which was used in Ang’s risk classification, had no statistical significance in PFS, DSS, and OS at all (data not shown).” I think this data needs to be included to compel the reader.

Response of the authors: According to the reviewer’s suggestion, supplementary Table is added as below. Correspondingly “(supplementary Table)” was added in line191.

  • Reviewer 1: On Line 181: “N-stage, dichotomized between N2a and N2b, as used in Ang’s risk classification, was statistically significant in OS and nearly significant in PFS and DSS with p-values of 0.072 and 0.051, respectively” please report the p-value for OS.

Response of the authors: According to the reviewer’s suggestion, p-value for OS was added in line 193. Additionally other significant p-values of PFS and OS were added in line 196.

  • Reviewer 1: On Line 298, the author states that “From the UVA and MVA, it appears quite difficult in this study to derive the risk classification with a combination of smoking amount and N-stage. It follows that Ang’s risk classification is not a reliable and robust prognostic factor in real-world HPV-related OPSCC of Japanese patients.” This is an exaggerated interpretation of the data. In the author’s results, N-stage and Ang’s risk did quite well in predicting outcomes in UVA, and N-stage did well in MVA. The authors set a strict criteria for “robust”, which was defined as “Ang’s risk classification can be considered robust, if the classification itself shows statistical significance both in UVA and MVA, and the 2 variables in the classification criteria (smoking amount and N-7) both are statistically significant in predicting prognosis as well.” There is also no mention of the author’s definition of “reliable”. I think a fairer assessment of the data is as follows. “While the smoking amount, N-stage, and Ang’s risk proved to be predictive of various combinations of outcomes within the UVA and MVA analyses, none met the criteria of consistently predicting each of PFS, DSS, and OS across UVA and MVA analyses. Thus, Ang’s risk does not meet our definition of robust and cannot necessarily be generalized and considered robust to all HPV-associated OPSCC patients who receive a variety of treatment modalities. Instead, Ang’s risk appears to be appropriate when certain patient characteristics and treatment conditions are met.”

Response of the authors: Thank you for your detailed suggestions. As the reviewer pointed out, the definition of robustness is very difficult and even we felt a little uneasy to make this definition of robustness. But at least MVA in this study repeatedly showed age would be more important than other variate, so that we used this criteria of robustness. And curiously, Ang’s risk was significant in some prognosis even the smoking habit was not significant. But if we accept this contradictory findings, Ang’s risk classification actually useful in some prognostic discrimination. Therefore we changed the sentence in line 298 “From the UVA and MVA, it appears quite difficult in this study to derive the risk classification with a combination of smoking amount and N-stage. It follows that Ang’s risk classification is not a reliable and robust prognostic factor in real-world HPV-related OPSCC of Japanese patients.” was changed as follows in line 314, “While the smoking amount, N-stage by TNM-7, and Ang’s risk classfication proved to be predictive of various combinations of outcomes within the UVA and MVA analyses, none met the criteria of consistently predicting each of PFS and OS across UVA and MVA analysis. Thus, Ang’s risk classification does not meet our definition of robustness and cannt necessarily be generalized and considered robust to all HPV-related OPSCC patients who receive a variety of treatment modalities. Instead, Ang’s risk classification appears to be appropriate when certain patient characteristics and treatment conditions are met.”  

  • Reviewer 1: On Line 304, the author states “However, even in the patients treated by definitive OP, the amount of smoking and N-stage dichotomy, both of which are criteria of Ang’s risk classification, were not unanimously significant in PFS, DSS, and OS. Therefore, Ang’s risk classification appears to be not so robust even in the patients treated by the definitive OP.” This sentence should be significantly altered. Just a few lines before, the author states “patients treated by the definitive OP, the low and intermediate risk patients by Ang’s risk classification showed statistically significant differences in PFS, DSS, and OS in the UVA, although a low number of patients precluded MVA.” Therefore, the author cannot claim that the Ang’s risk is not robust in the definitive OP group. Perhaps if there was enough power to run MVA, which then demonstrated Ang’s risk was not predictive of survival, the authors could argue the point. But as the data stands, it is not appropriate to do so. Lastly, I don’t think testing for “robustness” here is appropriate. Prior to this subgroup analysis, robustness referred to the entire patient cohort regardless of treatment modality. The authors should stick with this definition and use a different adjective to describe Ang’s risk in the scope of the definitive OP and definitive RT groups.

Response of the authors: Thank you for valuable and fruitful suggestions. As the reviewer suggested, “Therefore, Ang’s risk classification appears to be not so robust even in the patients treated by the definitive OP.” was deleted and replaced by “ Because of lack of the MVA, robustness of Ang’s risk classification in the patients treated by the definitive OP cannot be discussed. However, Ang’s risk classification appears to be useful in this patient cohort.”

General comments

  • Figures and Tables need to be cleaned up and look more professional. For example, remove grid lines from survival curves, increase font to be more legible, etc.

Response of the authors: Thank you for your instructions. Accordingly, we revised and improved quality of Figures and Tables.

  • Reviewer 1: Please change all instances of “real-world patients” to “the general HPV-related OPSCC patient population” or something analogous.

Response of the authors: According to the instruction of reviewer, all “real-world” was changed to “general”.

Line 45: “Ang’s risk classification was not so robust in predicting the prognosis of general Japanese HPV-related OPSCC patients.”

Line 66: “whether Ang’s risk classification is also valid in the general HPV-related OPSCC patient population treated with other modalities such as surgery, and a validity analysis has not been performed at all in the general Japanese patient population with HPV-related OPSCC.”

Line 304: “but its validity has not been confirmed in surgically treated patients and in the general Japanese patient population.”

Line 378: “Also, Ang’s risk classification turned out to be not a robust prognostic factor in the general Japanese patients with HPV-related OPSCC treated by various modalities including surgery.”

  • Reviewer 1: Please specify any potential sources of bias. For example, this was a single-institution study. It remains to be seen whether this trend holds across institutions with different surgical teams that treat Japanese OPSCC patients or different chemoradiation regimens.

Response of the authors: According to the suggestion of the reviewer, following sentences were added in Line 364 in the Discussion. “Furthermore, this study dealt with the patients treated in a single Japanese institution dedicating specifically to cancer therapy. It remains to be seen whether this trend holds across institutions with their own oncological teams treating the Japanese OPSCC patients using different surgical approaches and chemoradiotherapy regimens.”

  • Reviewer 1: In general, I believe the definition of “robust” needs to be altered. If the authors only mention the robustness of Ang’s risk in the title of the paper, then the definition of “robust” should only include the predictive power of Ang’s risk in UVA and MVA. I don’t think it is correct to include smoking amount and N-stage as additional criteria for “robust”- even if these variables go into the calculation of Ang’s risk. It is fine to note the discrepancy when Ang’s risk is predictive when smoking amount and N-stage are not. But to state that Ang’s risk is not robust if it does in fact predict outcomes in UVA and MVA doesn’t seem appropriate. Also, since Ang’s risk only considered PFS and OS (DSS was a secondary endpoint in the general RTOG-0129 trial), I don’t think DSS should be included as a criteria when trying to determine its validity.

Response of the authors: We appreciate deeply for your valuable suggestions. According to the reviewer’s suggestions, the definition of “robustness” was changed that Ang’s risk classification is robust when it showed statistical significance in PFS and OS by UVA and MVA, irrespective of the significance of smoking and N-stage. Line 144 “Ang’s risk classification can be considered robust, if the risks defined by the classification show statistical significant differences in PFS and OS by both UVA and MVA, since Ang et al. demonstrated the significance of their risk classification in PFS and OS [4].”

Reviewer 2 Report

The authors present an interesting manuscript focusing on HPV-related OPSCC and Ang´s et al prognostic algorithm.

However, there is some remarks and comments.

1) HPV-detection is based only on p16 IHC and this is pointed out in the Discussion as a limitation. Why the authors did not use an additive HPV-methodology for example HPV DNA PCR or mRNA ISH? Additional HPV detection method could make the analyses significantly more powerful.

2) The authors may consider to add number of patients to the Kaplan Meier curves below each time period.

3) Did the authors use backward or forward method for multivariate analyses? It seems now that all insignificant parameters in the univariate analyses are also included in the multivariate analyses.

Author Response

The authors deeply appreciate Reviewer 2 for the kind review of our manuscript, and insightful remarks and comments. We modified the manuscript according to the suggestions of Reviewer 2 as follows.

  • Reviewer 2: HPV-detection is based only on p16 IHC and this is pointed out in the Discussion as a limitation. Why the authors did not use an additive HPV-methodology for example HPV DNA PCR or mRNA ISH? Additional HPV detection method could make the analyses significantly more powerful.

 Response of the authors: Thank you for your comments. Because HPV DNA analysis and mRNA ISH studies are not covered by Insurance in Japan, so to study HPV DNA and mRNA ISH, we need an additional grant. Fortunately, one of our coauthors, Dr. Taisuke Mori, could obtain a grant for HPV DNA analysis and he will further extend the study. Accordingly, the authors added some comments about it in “limitation” from line 368, as follows,

“This study used p16 positivity as a surrogate for HPV infection and HPV-DNA and m-RNA analyses were not performed. There are numerous reports disclosing discrepancies between p16 positivity and HPV-DNA detection [31,39]. P16 positive but HPV-DNA negative patients show poorer prognosis in comparison to p16 positive and HPV-DNA positive patients. HPV-DNA analysis of the same patient cohort is planned and will be published in the near future”.

  • Reviewer 2: The authors may consider to add number of patients to the Kaplan Meier curves below each time period.

Response of the authors: According to the suggestion of the reviewer, numbers of the patients were added under Kaplan-Meier graphs (Figures 1-3).

  • Reviewer 2: Did the authors use backward or forward method for multivariate analyses? It seems now that all insignificant parameters in the univariate analyses are also included in the multivariate analyses.

Response of the authors: The authors did also forward methods in multivariate analysis, but in each survival, Ang’s risk classification was excluded and vanished during calculation because the p-value was not satisfactory. Therefore to make clear how large or small HRs were, HRs for all variables were presented with inputting all variables in proportional hazard model. I hope your understanding. 

Round 2

Reviewer 1 Report

I would like to thank the authors for thoughtfully addressing my comments from the initial review. There is now a good balance of discussing where Ang's risk was and was not successful and justification of why it does not meet the authors' definition of robustness overall. Each one of the comments was appropriately addressed, and thus I recommend this manuscript for publication following one very minor suggestion. 

The tables and survival plots are of sufficient quality now, but please add descriptive titles at the top of each of the survival plots for clarity. For example: "3-year PFS in the general Japanese population" or "Survival of patients treated with definitive operation". Also please add line color legends to the survival plots without them. 
